# Scoping Review of Current Costing Literature on Interventions to Reach Zero-Dose Children in Low- and Middle-Income Countries

**DOI:** 10.3390/vaccines12121431

**Published:** 2024-12-19

**Authors:** Ann Levin, Teemar Fisseha, Heidi W. Reynolds, Gustavo Corrêa, Tewodaj Mengistu, Nancy Vollmer

**Affiliations:** 1Levin & Morgan, LLC, Bethesda, MD 20817, USA; ann@levinmorgan.com; 2JSI Research & Training Institute, Inc., Arlington, VA 22202, USA; teemar_fisseha@jsi.com; 3Gavi, The Vaccine Alliance, 1211 Geneva, Switzerland; hreynolds@gavi.org (H.W.R.); gcorrea@gavi.org (G.C.); tmengistu@gavi.org (T.M.)

**Keywords:** immunization, cost, coverage increase, zero-dose, vaccination, LMICs

## Abstract

**Introduction:** A limited number of studies focus on estimating the costs of interventions to increase childhood immunization coverage in low- and middle-income countries (LMICs). Existing reviews often compare estimated costs but lack information on the methods used. The objective of this review is to summarize the methods used in costing studies that assessed interventions to reach zero-dose (ZD) children. **Methods:** We conducted a review of existing studies that estimate the costs of increasing childhood vaccination and reducing prevalence of ZD children in LMICs. We conducted searches of PubMed using terms including “immunization”, “cost”, “coverage increase”, “zero-dose”, and “LMIC”, and further extended our search to bibliographies and gray literature from organizations working to reach ZD children. We only included articles that estimated the cost of interventions to increase childhood vaccination and/or reach ZD children and not articles about introducing new vaccines or other age groups. We categorized each article according to their costing methods, cost components, types of costs calculated, and presence of uncertainty analysis. **Results:** Eleven articles met our inclusion criteria. Interventions costs varied from USD 0.08 per additional dose for SMS reminders in Kenya to USD 67 per dose for cash transfers in Nicaragua. Most of the studies were from South Asia: India (4), Pakistan (2), and Bangladesh (1). The rest were from Africa (3) and Latin America (1). Most articles did not include a description of their costing methods. Only three described their methods in detail. **Conclusions:** Few studies have estimated the costs of increasing childhood vaccination coverage and reducing the number of ZD children in LMICs. The wide variation in intervention costs underscores the need for standardized costing methodologies to enhance comparability across studies. Only three studies detailed their costing methods, making comparisons challenging. Establishing research principles for costing ZD interventions could strengthen future evidence for policymaking.

## 1. Introduction

The Gavi-funded Zero-Dose Learning Hub (ZDLH) is a global project led by JSI working to generate evidence and learning on interventions to identify and reach children that are either under-immunized or zero-dose (ZD) in low- and middle-income countries (LMICs). There are currently four Gavi-funded country learning hubs (Bangladesh, Uganda, Nigeria, and Mali) where the effectiveness of interventions to reach ZD children are being assessed and, in some cases, the costs are recorded and analyzed as well.

Zero-dose children are operationally defined by Gavi, the World Health Organization (WHO IA2030), and the ZDLH, as children who have not received the first dose of the diphtheria–tetanus–pertussis (DTP1)-containing vaccine by the end of their first year of life. While the concept of ZD interventions generally focuses on interventions that increase DTP1 coverage, the broader intent is to identify and reach children that have been missed during routine immunization (RI). Thus, there are various definitions of ZD interventions in the literature, ranging from interventions that increase childhood vaccination coverage to more specific ones such as reaching children that are under-vaccinated and have not received all of their required vaccines in the first year according to WHO recommendations. Interventions that are designed to increase immunization of ZD and under-immunized children often require additional resources and tailoring to the local context and can include a wide range of strategies, including periodic intensification of routine immunization, improvements in micro-planning, demand-side interventions, and systems improvement.

Despite over two decades of progress increasing access to childhood immunization, the number of ZD children in Gavi-supported countries has increased in recent years. ZD children and the communities where they live have been systematically missed by vaccination and other primary healthcare services and tend to suffer multiple forms of socioeconomic deprivation. Gavi is focused on achieving immunization equity and reaching ZD children through its current 5.0 strategy. The emphasis on equity and reaching ZD children requires differentiated, targeted subnational strategies that are tailored to the context. As a result, there is growing interest in understanding how to reliably measure the incremental cost of reaching ZD and under-immunized children as well as the cost-effectiveness of different ZD interventions.

Understanding of the costs of tailored delivery strategies is important, as costs may vary widely across the types of interventions needed to address the immunization barriers in different ZD contexts across countries and populations. Many immunization costing studies have been conducted over the past four decades. These studies have various objectives: (1) informing decision-making regarding new vaccine introduction [1]; (2) evaluating the efficiency and/or effectiveness of immunization programs at different levels [2]; (3) conducting planning and advocacy for resource mobilization [3]; (4) evaluating the cost-effectiveness of different strategies such as routine vs. supplementary immunization activities (SIAs) or for elimination of diseases [4]; and (5) estimating the cost of increasing vaccination coverage [5,6]. Estimating the costs of increasing coverage and reaching ZD children is particularly important because of the challenges associated with reaching marginalized populations and other hard-to-reach groups. Information about the costs of different types of ZD interventions is needed for planning purposes, to advocate for funding with policymakers and program managers, and to evaluate the cost-effectiveness of different interventions. Information on cost-effectiveness can be used in decision-making to improve resource use, i.e., allocative efficiency. It is also important to conduct these studies to ascertain whether it is more costly to reach ZD children and in which contexts.

Costing studies tend to differ in their methods, which makes it difficult to develop a general estimate of the additional costs of reaching ZD and reduces the comparability across studies. Some of the differences are related to whether ingredients or top–down costing is used, the type of sampling employed, the comprehensiveness of cost categories, and types of costs analyzed (e.g., financial and economic, recurrent and capital, startup and implementation). Cost estimates will be more rigorous if a more disaggregated methodology, such as ingredients costing, is used and both startup and implementation costs are included, as this approach increases the likelihood of capturing all resources employed. Analyzing economic costs as well as financial costs is also useful, since it includes opportunity costs (e.g., in-kind and donated items) as well as accounting costs (monetary costs incurred).

This scoping review provides a synthesis of articles that estimated the costs of interventions to reach ZD children using primary data as opposed to modeled estimates. This review differs from earlier reviews since it focuses on methods used in costing of ZD interventions as well as their findings.

## 2. Methods

Our search strategy aimed to identify peer-reviewed articles and gray literature published after 2005 that reviewed or estimated the costs of interventions targeting ZD or hard-to-reach children, with a focus on increasing overall vaccination coverage in LMICs. We broadened our search to include articles on the costs of increasing childhood immunization coverage rather than focusing specifically on DTP1, as the latter definition was rarely used in studies. In addition, we searched for studies that focused on increasing childhood immunization coverage rather than reducing ZD, since the ZD terminology is relatively new and few studies focus on it. Limiting the review to studies published post-2005 allowed us to build upon earlier systematic reviews without duplicating prior findings. The methods are compliant with PRISMA guidelines for a scoping review.

Our methodology comprised three main approaches:We searched PubMed using the following search terms: immunization, cost, coverage increase, zero-dose, vaccination, LMICs.We reviewed the bibliographies of systematic review articles to identify studies published after 2005 that focused on LMICs and included cost analyses of increasing childhood vaccination.We searched gray literature from organizations that are implementing projects intended to reach ZD children in LMICs, such as Thinkwell, JSI, and International Vaccine Access Center (IVAC), and conducted additional searches on Google.

The inclusion criteria were studies that had the following characteristics: (1) focused on interventions that increased childhood vaccination, (2) estimated the costs of the interventions, (3) were from LMICs, and (4) were conducted after 2005.

We selected search terms pertaining to immunization, cost, and increasing coverage. We also included vaccination, LMIC, and zero-dose in our search terms. Our search included papers on increasing child vaccination as well as reducing ZD children, since the focus on ZD is relatively recent.

For each selected paper that estimated the cost of interventions to increase coverage, we reviewed and documented information on the study context; the methods used, including incremental and/or retrospective methods; financial and economic cost; cost components; and output/outcome variables. The study findings on cost per dose or per child were also documented.
Key Concepts and Terms Used in this Paper**Cost per additional dose for intervention**: Additional intervention cost of providing another dose (calculated by dividing the extra cost by additional vaccine doses administered).**Cost per intervention**: Total cost of implementing an intervention (calculated by adding the value of the cost categories together).**Cost per percentage change in immunization coverage**: Cost divided by additional percent change in immunization coverage.**Cost per person reached**: Intervention cost divided by number of persons reached.**Incremental cost-effectiveness ratio (ICER)**: Measure that summarizes the cost-effectiveness of an intervention (ratio of costs of intervention divided by effectiveness measure—such as deaths or disability-adjusted life years (DALYs) averted). **Zero-dose child:** Child that has not received the first dose of the DTP-containing vaccine.

## 3. Results

### 3.1. Reviews

We found three systematic reviews that focused on estimating the costs of increasing childhood immunization coverage after 2005. Two of the papers reviewed articles in high-income countries as well as LMICs. Munk’s review [5] focused on studies in LMICs. Table 1 summarizes the findings of the three systematic reviews.

Munk et al. [5] found 14 articles that included data on the costs of increasing childhood vaccination coverage in LMICs. They found that interventions aimed at improving delivery mechanisms were associated with the largest coverage increases. The intervention cost per person in 2017 ranged from USD 1.38–USD 162.25, with a median value of USD 23.64. The authors found that many of the included studies did not describe the costing methods used in their analyses. Seven of the fourteen articles in Munk’s systematic review also met our criteria of the current synthesis, i.e., estimated costs of increasing coverage of childhood vaccination after 2005. Articles were excluded if they focused on vaccines targeting populations over two years of age, did not provide the cost per dose or cost per person, or did not estimate costs (the study authors contacted some of the individual authors to obtain cost data).

Yemeke et al. [8] found 30 articles that reported on the estimated cost of increasing immunization coverage in LMICs. The mean cost per dose of interventions was USD 0.41 in low-income countries and USD 18.86 in middle-income countries. Only five articles from the Yemeke systematic review met our study criteria, i.e., focused on the costs of interventions to increase immunization uptake rather than costs of introducing vaccines or costs of vaccines that targeted older age groups (e.g., oral cholera vaccine and HPV), or did not report on the costs.

Ozawa et al. [7] found 42 articles, of which 17 focused on increasing immunization coverage in LMICs. The study estimated that the intervention cost per 1% increase in coverage (the authors estimated the increase in immunization coverage that occurred after the intervention was introduced and then calculated the cost of increasing coverage by 1%) was USD 0.06 in low-income countries and that the cost increases with baseline immunization coverage level, reflecting the higher cost of reaching under-vaccinated children. This study differs from the other two reviews, since the authors estimated the intervention cost per dose per percent coverage change compared to baseline coverage. It should be noted, however, that the estimates in this cost function combine intervention costs per dose from diverse interventions and these may not be comparable. Six of the articles included in Ozawa [7] meet our study criteria for the current synthesis.

### 3.2. Search on PubMed

Most of the articles for this study had already been identified in the three previous systematic reviews or did not meet our study inclusion criteria. We found one additional article that was not identified in the three reviews—a study by Chatterjee et al. [6] that focused on estimating the costs of increasing coverage through the Intensified Mission Indradhanush (IMI) Program in India.

### 3.3. Gray Literature

We identified three articles that examined economic issues related to ZD children [9,10,11]. Two of the papers focused on the economic benefits of immunization for ZD children and the third paper focused on the insufficiency of funding to reach ZD children. These papers did not focus on the costs of vaccination interventions and therefore, all three were excluded from the review. Figure 1 presents a PRISMA flow diagram of our search strategy.

### 3.4. Articles on Increasing Coverage of Childhood Vaccination in LMICs

Eleven (note that one additional study [12] utilized Chatterjee’s estimates to estimate cost per ZD child reached through the IMI program in India) peer-reviewed articles met our criteria for inclusion: studies that estimated the cost of interventions to increase childhood vaccination coverage from an LMIC. We categorized each article that met the criteria according to the following: descriptions of their costing methods, tables or figures showing the costs by component, the type of costs calculated, and whether uncertainty analysis was conducted. Table 2 provides detailed information about the reviewed studies, including the study methods.

The interventions in the included studies differ in their approaches to increasing childhood vaccination. Three of the studies focus on the education of mothers/caregivers on the benefits of immunization [13,14,15]; three studies focus on reminding caregivers to take their children for vaccination [16,17,18]; three papers examine interventions that identify vaccination defaulters and conduct follow-up vaccination [6,19,20]; one article focuses on immunization camps in rural areas [21]; and one article focuses on cash transfers that require children to attend preventative health visits [22].

The costs of the interventions range from USD 0.08 per additional dose for a cellular phone contact to encourage return visits in Kenya [18] to USD 84 for an intervention of immunization camp and incentive intervention in rural villages in India [21]. The median intervention cost per additional dose is USD 3.72 (see Table 2).

**Table 2 vaccines-12-01431-t002:** List of articles that estimate costs of increasing childhood vaccination coverage in LMICs after 2005.

Author and Journal	Country	Article Name	Description	Outcome of Interest	Intervention Cost (Cost per Dose/Child) (2017 USD)	Intervention Cost per ZD Child Reached	Costing Methods
Andersson [14] *BMC Int Health and Human Rights*	Pakistan	Evidence-based discussion increases childhood vaccination uptake: a randomized cluster-controlled trial of knowledge translation in Pakistan	Costing of structured group discussion education intervention on vaccination benefits with adults in rural communities with low vaccination uptake	DTP3 and measles vaccination status	Estimated government intervention cost was USD 90,000 total (USD 12.63 per dose)	NA	Cost components not described. Notes that the cost of provincial and national coordinators working on the project were excluded.
Banerjee [21] *BMJ*	India	Improving immunization coverage in rural India: clustered randomized controlled evaluation of immunization campaigns with and without incentives	Costing of an immunization camp and incentives intervention in rural villages	Proportion of children aged 1–3 that were partially or fully immunized	USD 41,109 vs. USD 27,420 (USD 84 per child vs. USD 42 per child in 2017 USD)	NA	Methods described and cost components Includes worker salaries, travel, village worker honorarium, training, refrigerators, monitoring, and incentives.
Bangure *BMC Public Health* [16]	Zimbabwe	Effectiveness of short message services reminders on a childhood immunization program in Kadoma, Zimbabwe—a randomized controlled trial	Effectiveness of a short message services (SMS) reminder on a childhood immunization program in Kadoma, Zimbabwe	Pentavalent Immunization coverage at 6, 10, and 14 weeks	NA (USD 0.0.25)	NA	Costs include messages to study participants (USD 57.46) and capturing of data cost (USD 0.33 per message for human resources).
Barham [22] SSRN Electron J.	Mexico and Nicaragua	Beyond 80%: Are there new ways of increasing vaccination coverage?	Cash transfers conditional on children attending preventative health visits and mothers attending health education talks	DTP3 coverage	Mexico: USD 2.3 billion total (USD 44 per child 2017 USD) Nicaragua: USD 5,000,000 (USD 67 per child 2017 USD)	NA	Costs include program and transfer costs.
Chatterjee et al. [6] *Health Policy and Planning*	India	The incremental cost of improving immunization coverage in India through the Intensified Mission Indradhanush Program	Identified missed children and vaccinated them in temporary outreach sites for one week over a consecutive 4-month period	Full immunization coverage	No cost for program as a whole (USD 3.20 per dose) in Uttar Pradesh and highest in Maharashtra (USD 11.35 per dose)	USD 82.88 *	Costs estimated economic and financial incremental costs; Data collected at district and sub-district levels.
Haji [17] *BMC Public Health*	Kenya	Reducing routine vaccination dropout rates: evaluating two interventions in three Kenyan districts	Costing of text message reminder intervention in three districts	Pentavalent3 vaccination	USD 136 (USD 0.33 per child)	NA	Costs include 1488 messages sent to the participants in the SMS, and premium cost of scheduling messages from web for six months (USD 66.70).
Hayford [19] *Vaccine*	Bangladesh	Cost and sustainability of a successful package of interventions to improve vaccination coverage for children in urban slums of Bangladesh	Extended hours at satellite clinics; training; clinic screening tool; volunteer community group	DTP3, full immunization coverage	NA (USD 2.84 per dose; USD 22.15 per fully immunized child	NA	Used document review and stakeholder interviews to collect data. Estimated both financial and opportunity costs of the intervention, including uncompensated time, training and supervision costs.
Mokaya [18] *Pan African Medical Journal*	Kenya	Use of cellular phone contacts to increase return rates for immunization services in Kenya	Costing of a cellphone contact intervention to track defaulters	Pentavalent2, Pentavalent3 status	Aver. cost tracking w/phone (USD 0.07 per child)	NA	Used the monthly post-bills to confirm the cost of the calls and multiplied by the frequency.
Owais [13] *BMC Public Health*	Pakistan	Does improving maternal knowledge of vaccines impact childhood immunization rates?	Three targeted pictorial messages regarding vaccines	DTP3 completion	NA (USD 1.15 per child)	NA	Cost of laminated colored pictorial cards used by CHWs as well as pamphlets; Estimate that cost of national scale-up will be USD 200,000–USD 100,000 for cards and pamphlets and USD 100,000 for training sessions.
Powell-Jackson [15] *PLoS Med*	India	Effect and cost-effectiveness of educating mothers childhood DPT vaccination on immunization uptake, knowledge and perceptions in Uttar Pradesh: A randomized controlled trial	Intervention on educating mothers about childhood DTP vaccination	Proportion of children that received DTP3 vaccination	USD 11,137 (USD 24 per child)	NA	Costs collected through accounting systems and categorized as startup/implementation; estimated cost of scale-up assuming no change to average cost but double the children reached.
Rainey [20]	India	Providing monovalent oral polio vaccine type 1 to newborns	Identifying and vaccinating newborns with OPV within 72 h of birth	OPV1 vaccination	NA (USD 3.72 per dose)	NA	Cost of motorbikes, per diem, travel allowance, incentives, and delivery kits for traditional birth attendants. Does not include table listing cost components/drivers.

* Estimated in [12]. Figure 2 illustrates the median intervention cost per dose per person for each type of intervention. For the three types of interventions—reminders to caregivers, identification and vaccination of defaulters and maternal education on the benefits of vacation—the median cost was under USD 15 per person per dose. In contrast, immunization camps and cash transfers interventions showed higher median costs, ranging from USD 40–55 per person, making them at least 400% more expensive than the other interventions. It is important to note that these latter interventions provide additional preventative health services beyond immunization, which makes direct comparisons with the first three types of interventions challenging.

### 3.5. Description of Costing Methods

Only three of the papers provided detailed descriptions of their costing methods [6,19,21]. Other papers mentioned that they calculated costs but did not go into detail about the cost components and methods of analysis.

Table 3 shows the methods used in these papers. All three papers described their methods and included tables and figures with cost components and percentages of total costs. Two of the three conducted uncertainty analyses to account for uncertainty in the cost values with the use of sensitivity analyses or confidence intervals.

## 4. Discussion

This review highlights the scarcity of studies that estimate the costs of interventions aimed at increasing childhood vaccination or reaching ZD in LMICs, as most studies evaluating vaccination interventions in these contexts do not provide detailed cost estimates. The eleven studies included in this review assessed different types of interventions and estimated the cost of increasing childhood vaccination but did not specifically address the cost of reaching ZD children. Median costs varied widely across interventions, from USD 0.30 per person for SMS reminders to caregivers to USD 56 per person for cash transfer programs.

Three studies examined interventions that cost less than USD 15 per person, suggesting that some strategies may be affordable for governments. However, it should be noted that costs and affordability are context-specific and sometimes depend on the baseline coverage levels. For example, the cost of a universal cash transfer program aimed at increasing coverage (in an area that is already close to 100%) cannot be compared to the cost of targeted demand incentive approaches in communities with existing low baseline levels. In addition, the costs may differ substantially depending on the costing methods used.

Geographically, seven of the included studies were from South Asia (India, Pakistan and Bangladesh); three were from African countries; and the remaining study was from Latin America.

Only three papers provided detailed descriptions of their cost estimation methods, including the types of costs estimated (e.g., recurrent and capital costs, or financial and economic costs). Two of the three papers included some analysis of uncertainty in their cost estimates.

Relatively few studies on the cost of reaching ZD children or increasing childhood immunization coverage in LMICs are available. Most studies have focused more on reaching higher DTP3 coverage rather than the reduction in the percentage of ZD children. The paucity of studies may partially stem from the complexity of costing programmatic interventions that target ZD children, as calculating the cost of increasing coverage is often challenging. Defining pre-intervention costs and allocating costs within multi-intervention projects is difficult, especially when some activities may pre-date the project itself. Relatively few of the studies that do exist describe the cost estimation methods used, which complicates efforts to assess the adequacy of their analyses and comparability to other studies.

### Limitations

Comparing studies that estimate the cost of increasing vaccination coverage is difficult due to differences in measured outcomes (e.g., cost per child/person exposed vs. cost per dose vs. cost per intervention). Additionally, variations in intervention design make direct comparisons difficult, as each program may use distinct approaches and target populations. Local and regional differences in costs for different categories (e.g., labor and commodities) can also affect the ability to make comparisons among studies.

## 5. Conclusions

In conclusion, the current literature on the costs of reaching ZD children in LMICs is insufficient to develop a solid evidence base for decision-making. Our review shows that per-dose costs vary widely by intervention type, with cash transfer programs being among the most expensive. However, most studies that have estimated the costs of increasing coverage have not adequately described their methods. Only three studies described the methods that they used to estimate costs. Thus, it is difficult to compare the costs among studies. More information on the intervention costs is needed to provide important insight for understanding the resources incurred for the purposes of improving implementation, planning, and as a component for cost-effectiveness studies.

Ideally, cost studies should be incentivized by global funding agencies whenever an immunization equity approach is implemented. Future studies should prioritize clear and standardized descriptions of costing methods for ZD interventions through referring to guidance documents on immunization costing [2,23,24]. It is also important to prioritize the generation of costing evidence from subnational areas to generate reliable costing data for different zero-dose contexts. Additionally, the national and subnational settings should be described in all costing studies to improve comparability across studies and contexts. Where possible, costing analyses should estimate the additional costs associated with identifying, planning and reaching ZD and under-vaccinated children through tailored delivery strategies. Establishing principles for cost estimation and standardized cost components would strengthen the comparability of ZD intervention studies, allowing for findings to be synthesized across populations and countries. Costing studies aligned with these principles would offer a more comparable evidence base to inform resource allocation and policy decisions aimed at achieving universal immunization coverage in LMICs.

## Figures and Tables

**Figure 1 vaccines-12-01431-f001:**
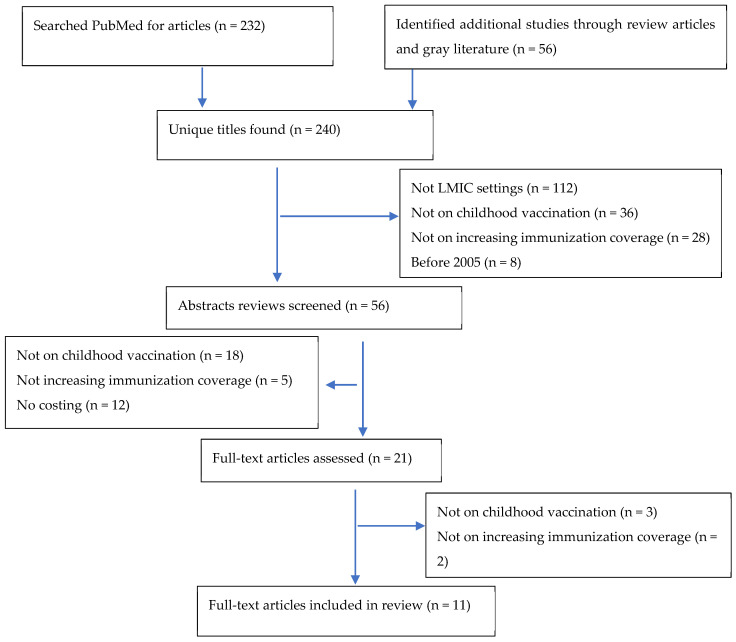
PRISMA flow diagram.

**Figure 2 vaccines-12-01431-f002:**
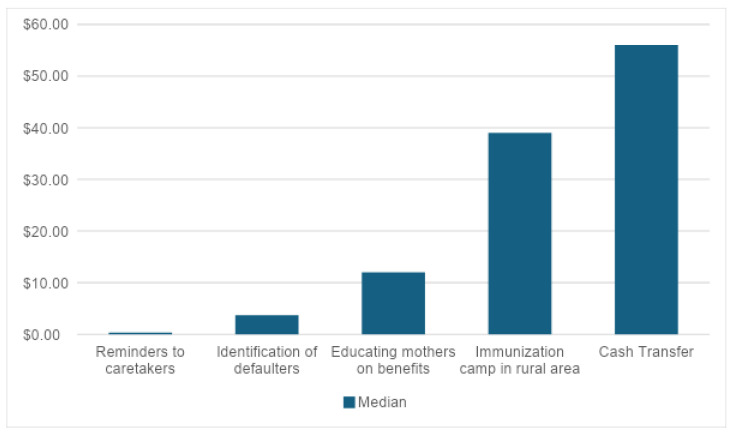
Median intervention cost per dose/person for increasing immunization coverage (2017 USD).

**Table 1 vaccines-12-01431-t001:** Description of systematic reviews on estimation of costs of increasing immunization doses delivered in LMICs.

Authors	Year	Title	Findings
Munk, et al. [5]; BMC Health Services Research https://doi.org/10.1186/s12913-019-4468-4	2019	Systematic review of the costs and effectiveness of interventions to increase childhood vaccination coverage in LMICs	14 articles on the costs of increasing coverage with considerable heterogeneity in costing methods. Incremental cost-effectiveness ratios (ICERs) ranged from USD 0.66 to USD 162 per child reached with a dose; vaccines included DTP3 and measles.
Ozawa, et al. [7] Vaccine https://doi.org/10.1016/j.vaccine.2018.05.030	2018	Systematic review of the incremental costs of interventions that increase immunization coverage	42 articles met criteria, 17 of which were in LMICss; intervention cost per dose per percent change in coverage ranged from USD 0.01 to USD 38.16 [mean USD 3.13, SD USD 7.02] for various interventions to increase immunization coverage.
Yemeke, et al. [8] Vaccine https://doi.org/10.1016/j.vaccine.2021.05.075	2021	Promoting, seeking, and reaching vaccination services: A systematic review	57 studies describing information, education and communication (IEC) costs, social mobilization costs, and costs of interventions to increase vaccination demand, 30 of which were in LMICs: mean costs per dose in low-income countries were USD 0.41 (SD USD 0.83) and in middle-income countries they were USD 18.87 (USD 50.65); vaccines included Hepatitis B, meningitis, HPV, measles, and oral cholera.

**Table 3 vaccines-12-01431-t003:** Costing methods used in three studies with detailed description.

	Describes Costing Methods	Displays Cost Components with Percentiles	Calculates Economic Costs	Defines Other Types of Costs Used (Recurrent, Fixed)	Uses Uncertainty Analysis
Banerjee [21]	In-study appendix	Yes	No	Fixed, average, and marginal costs	Sensitivity analysis
Chatterjee [6]	Yes	Yes	Yes	Incremental costs	Calculation of confidence intervals
Hayford [19]	Combination of activity-based costing and ‘ingredients’ approach	Yes	Yes	Recurrent costs	Not available

## Data Availability

Not applicable.

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
