# Peer review of "Scoping Review of Current Costing Literature on Interventions to Reach Zero-Dose Children in Low- and Middle-Income Countries"

_vaccines, 2024, doi:10.3390/vaccines12121431_

Round 1

Reviewer 1 Report

Comments and Suggestions for Authors

In this manuscript, the authors conducted searches in PubMed and found the wide variation of intervention cost based on limited previous reports and also the scarcity of literatures with detailed method description. Therefore, to facilitate the interventions to reach zero-dose children in low- and middle-income countries, the authors emphasized the importance to establish research principles and to standardize the costing methodology in the future.

The manuscript is well written, and the conclusion that the current literatures are insufficient to develop a solid evidence base for decision-making clearly point out the direction for future studies

Author Response

Comments 1: In this manuscript, the authors conducted searches in PubMed and found the wide variation of intervention cost based on limited previous reports and also the scarcity of literatures with detailed method description. Therefore, to facilitate the interventions to reach zero-dose children in low- and middle-income countries, the authors emphasized the importance to establish research principles and to standardize the costing methodology in the future.

The manuscript is well written, and the conclusion that the current literatures are insufficient to develop a solid evidence base for decision-making clearly point out the direction for future studies

Response 1: Thank you for the positive feedback, this is well noted.

Reviewer 2 Report

Comments and Suggestions for Authors

The manuscript entitled “Review of Current Costing Literature on Interventions to Reach Zero-Dose Children in Low- and Middle-Income Countries” addresses the important issue of interventions aimed at increasing childhood vaccination and reaching zero-dose children in low- and middle-income countries (LMICs). This is a highly relevant and timely topic, considering global efforts to increase immunization coverage and reduce preventable diseases among underserved populations. The authors conducted a thorough review of existing literature on the costs associated with various immunization interventions in LMICs. They included a wide range of studies and costing methods, which provides a robust foundation for understanding cost variability across different intervention types and contexts. The results section is well-organized, categorizing studies based on intervention types, cost estimates, and geographical regions. The use of tables and figures enhances the clarity of the information, making it easier for readers to compare cost data across studies. The manuscript effectively identifies gaps in the current literature, such as the lack of standardized costing methods and limited focus on zero-dose children specifically. This highlights the need for further research and standardized methodologies in the field.

However, these aspects should be improved:

1.      While the manuscript mentions the costing methods used in some studies, it would benefit from a more in-depth explanation of how different costing methods impact cost estimates. A dedicated section comparing the strengths and weaknesses of each method would provide valuable insights into why costs vary so widely.

2.      Although the manuscript reviews studies on childhood vaccination costs, it does not consistently distinguish between studies that focus on general immunization interventions versus those explicitly aimed at reaching zero-dose children. A clearer distinction throughout the text would strengthen the manuscript’s focus on zero-dose interventions.

3.      The discussion section could be expanded to include more specific limitations of the reviewed studies, such as variations in intervention definitions and regional differences. Additionally, more concrete recommendations for future research, such as establishing standardized cost components or conducting subnational costing analyses, would enhance the manuscript’s practical applicability.

4.      Some terms, like "cost per additional dose" or "cost per intervention," could be further clarified to ensure readers fully understand how these metrics are calculated and interpreted. Including a glossary or definitions table might be helpful, especially for readers less familiar with economic evaluations.

5.      While the manuscript discusses the importance of understanding costs, it could benefit from an expanded analysis of how these cost findings might inform policy decisions. For example, which types of interventions are most cost-effective for scaling up? How can policymakers use this data to allocate resources more efficiently?

The manuscript provides a solid foundation on the costs of immunization interventions in LMICs. Addressing the above areas for improvement would enhance its clarity, applicability, and depth, making it a more valuable resource for researchers and policymakers alike.

Author Response

Comments 1: The manuscript entitled “Review of Current Costing Literature on Interventions to Reach Zero-Dose Children in Low- and Middle-Income Countries” addresses the important issue of interventions aimed at increasing childhood vaccination and reaching zero-dose children in low- and middle-income countries (LMICs). This is a highly relevant and timely topic, considering global efforts to increase immunization coverage and reduce preventable diseases among underserved populations. The authors conducted a thorough review of existing literature on the costs associated with various immunization interventions in LMICs. They included a wide range of studies and costing methods, which provides a robust foundation for understanding cost variability across different intervention types and contexts. The results section is well-organized, categorizing studies based on intervention types, cost estimates, and geographical regions. The use of tables and figures enhances the clarity of the information, making it easier for readers to compare cost data across studies. The manuscript effectively identifies gaps in the current literature, such as the lack of standardized costing methods and limited focus on zero-dose children specifically. This highlights the need for further research and standardized methodologies in the field.

Response 1: Thank you for this feedback, this is well noted.

Comments 2: While the manuscript mentions the costing methods used in some studies, it would benefit from a more in-depth explanation of how different costing methods impact cost estimates. A dedicated section comparing the strengths and weaknesses of each method would provide valuable insights into why costs vary so widely.

Response 2: We added paragraph explaining why methods that use disaggregated costs are preferable since these improve the rigor of the study.

Comments 3: Although the manuscript reviews studies on childhood vaccination costs, it does not consistently distinguish between studies that focus on general immunization interventions versus those explicitly aimed at reaching zero-dose children. A clearer distinction throughout the text would strengthen the manuscript’s focus on zero-dose interventions.

Response 3: We added a sentence to explain why we included studies that focused on increasing immunization coverage rather than reducing ZD because of the infrequent usage of the term ‘ZD’ in the literature.

Comments 4: The discussion section could be expanded to include more specific limitations of the reviewed studies, such as variations in intervention definitions and regional differences. Additionally, more concrete recommendations for future research, such as establishing standardized cost components or conducting subnational costing analyses, would enhance the manuscript’s practical applicability.

Response 4: The discussion section was expanded with additional limitations and recommendations.

Comments 5: Some terms, like "cost per additional dose" or "cost per intervention," could be further clarified to ensure readers fully understand how these metrics are calculated and interpreted. Including a glossary or definitions table might be helpful, especially for readers less familiar with economic evaluations. 

Response 5: We added to the definitions in the Box of Key Concepts for ‘cost per additional dose’ and ‘cost per intervention’.

Comments 6: While the manuscript discusses the importance of understanding costs, it could benefit from an expanded analysis of how these cost findings might inform policy decisions. For example, which types of interventions are most cost-effective for scaling up? How can policymakers use this data to allocate resources more efficiently?

Response 6: We added a sentence on the use of data on cost-effectiveness of interventions.

Reviewer 3 Report

Comments and Suggestions for Authors

Thanks for providing this interesting literature review on the costs of Interventions to Reach Zero-Dose Children in Low- and Middle-Income Countries. 

If the article could be considered for publication, I'd suggest starting with attempting to measure in LMIC what is size of the problem in these countries, that is not mentioned yet. The latter is to understand better the need of having these interventions even before estimating if those could be affordable.  There should be a clearer section on the type of costing methodologies that had been used by multiple authors to estimate costs, the current authors put a lot of emphasis suggesting there is no data and very few studies had included their costs methods, but this could be considered as a weak conclusion for a paper to be published at Vaccines Journal.

In the discussion section, I'd suggest authors to suggest or recommend ways the costing should be generated to estimate Interventions costs to Reach Zero-Dose Children in Low- and Middle-Income Countries. Always leaving a positive conclusion could be useful for further readers and researchers to explore more in the field and try to test best ways to adopt costing methodologies, either used as best practices or new proposals.

Comments on the Quality of English Language

The quality of English Language is ok from my perspective.

Author Response

Comments 1: Thanks for providing this interesting literature review on the costs of Interventions to Reach Zero-Dose Children in Low- and Middle-Income Countries. 

If the article could be considered for publication, I'd suggest starting with attempting to measure in LMIC what is size of the problem in these countries, that is not mentioned yet. The latter is to understand better the need of having these interventions even before estimating if those could be affordable. 

Response 1: Thank you for this comment. We added relevant content to the introduction section in response to your suggestion.

Comments 2: There should be a clearer section on the type of costing methodologies that had been used by multiple authors to estimate costs, the current authors put a lot of emphasis suggesting there is no data and very few studies had included their costs methods, but this could be considered as a weak conclusion for a paper to be published at Vaccines Journal.

Response 2: Most of the authors did not describe the costing methodologies that they used. However, the three articles that describe their methods also described the type of costs that they estimated. Their methods are described in Table 3. Also, we added more detail on the methods to Table 3 in response to your suggestion.

Comments 3: In the discussion section, I'd suggest authors to suggest or recommend ways the costing should be generated to estimate Interventions costs to Reach Zero-Dose Children in Low- and Middle-Income Countries. Always leaving a positive conclusion could be useful for further readers and researchers to explore more in the field and try to test best ways to adopt costing methodologies, either used as best practices or new proposals.

Response 3: We added to the discussion section and suggested that future studies should refer to available guidance documents that have standardized descriptions of costing methods – e.g. Levin 2022, Brenzel 2015, Resch 2020.

Reviewer 4 Report

Comments and Suggestions for Authors

I sincerely believe that the work is very relevant. As discussed in the project 14 of the WHO program, each country, in addition to the neonatal or child mortality rates under 5 years, should record the number of people (including children) vaccinated in accordance with national immunization programs. This study tries to shed light on the difficult issue of unvaccinated children. Perhaps it would have been worthwhile to try to find more detailed other information from each country cited in the article to assess the potential reliability of the obtained data on z-children. and then calculate the economic burden. but perhaps the authors will do this in the continuation of their study

Author Response

Comments 1: I sincerely believe that the work is very relevant. As discussed in the project 14 of the WHO program, each country, in addition to the neonatal or child mortality rates under 5 years, should record the number of people (including children) vaccinated in accordance with national immunization programs. This study tries to shed light on the difficult issue of unvaccinated children. Perhaps it would have been worthwhile to try to find more detailed other information from each country cited in the article to assess the potential reliability of the obtained data on z-children. and then calculate the economic burden. but perhaps the authors will do this in the continuation of their study.

Response 1: Thank you for this comment. Unfortunately, the articles in our review do not provide sufficient information for us to assess the reliability of the data on z-children or calculation the economic burden. 

Round 2

Reviewer 3 Report

Comments and Suggestions for Authors

No additional comments from my end.